# Teriparatide as an Effective Nonsurgical Treatment for a Patient with Basicervical Peritrochanteric Fracture Nonunion—A Case Report

**DOI:** 10.3390/medicina58080983

**Published:** 2022-07-23

**Authors:** Cheng-Han Ho, Shi-Chien Tzeng, Chui-Jia Farn, Chia-Che Lee

**Affiliations:** 1Department of Orthopedic Surgery, National Taiwan University Hospital No.7, Zhongshan S. Rd., Zhongzheng Dist., Taipei 100225, Taiwan; durantfive@gmail.com (C.-H.H.); irene.lover@gmail.com (S.-C.T.); cjfarn@gmail.com (C.-J.F.); 2Graduate Institute of Biomedical Electronics and Bioinformatics, National Taiwan University, Taipei 100225, Taiwan

**Keywords:** teriparatide, nonunion, peritrochanteric fracture

## Abstract

The nonunion rate of surgically treated basicervical peritrochanteric fractures has been reported to be as high as 9%. Due to the high 1-year mortality rate following revision surgery, finding an effective nonsurgical treatment option is of interest. Over the last decade, numerous reports have been published that have suggested teriparatide as an effective treatment for certain types of fracture nonunion. However, the literature focused on teriparatide treatment for proximal femoral fracture nonunion is scanty. A 70-year-old man suffering from a left hip basicervical peritrochanteric fracture received cephalomedullary nail fixation. Nine months after the surgery, the patient still complained of left hip pain referring to the medial thigh with an antalgic limping gait. No sign of healing was noted for more than a consecutive 3 months of follow-up. Fracture nonunion was diagnosed and further confirmed by the computed tomography (CT). The patient preferred nonsurgical treatment after thorough discussion. He then received 4 months of subcutaneous teriparatide injections, 20 mcg daily. After less than 4 months of teriparatide treatment, a follow-up CT confirmed fracture union and the patient’s pain subsided. The patient also tolerated independent ambulation afterward. Teriparatide has been reported to be an effective treatment for certain types of fracture nonunion. Our case goes a step further to expand its possible application for basicervical peritrochanteric fracture nonunion. However, further larger scale studies are needed to confirm its efficacy.

## 1. Introduction

Basicervical peritrochanteric fractures are relatively rare injuries and account for only 1.8% of all proximal femoral fractures [1]. Previous studies have shown that the surgical complication rate is associated with numerous factors, including fracture morphology, reduction quality, tip-apex distance and the selected implants [2,3,4]. Nonunion rates were reported to be as high as 9% [5]. Various surgical or nonsurgical treatment options exist for the management of fracture nonunion. For elderly patients, however, revision surgical intervention may not be tolerable and the 1-year mortality rate following revision fixation has been reported to be up to 30% [4]. For those patients with a higher mortality risk following surgery, an effective conservative treatment may be an attractive treatment option. Over the last decade, some evidence has suggested that teriparatide not only treats osteoporosis but also promotes fracture healing in certain types of fracture nonunion [6]. However, the evidence about its use in basicervical fracture nonunion remains scarce. Herein, we describe a case of an elderly male, who had fracture nonunion of a left hip basicervical fracture for more than 9 months after being surgically treated with cephalomedullary fixation. Union was achieved after the patient had teriparatide treatment for less than 4 months.

## 2. Case Report

A 70-year-old male with a history of hypertension and diabetes mellitus was sent to our emergency department for severe left hip pain after being hit by a vehicle as a pedestrian. The radiographs of the pelvis and left hip revealed a left basicervical peritrochanteric fracture with displacement, femoral neck comminution and varus angulation (Figure 1). After the preoperative evaluation, he was brought to the operating room the second day after arrival and underwent closed reduction and internal fixation with an anti-rotation cephalomedullary device.

On day 2, the postoperative radiographs showed some varus alignment and neck shortening due to fracture comminution (Figure 2). The rehabilitation protocol was initiated during admission, including ambulation with weight bearing as tolerated. The patient was discharged without other perioperative complications.

During serial outpatient clinic follow-ups, the radiographs showed gradual neck shortening as expected due to the selected implant and fracture comminution. The bone mineral density examination was conducted, and osteopenia was diagnosed, with a T-score = −2.4 over the right total hip region. The patient received instruction for adequate daily calcium and vitamin D supplements. Nine months after the operation, the patient still complained of left hip pain referring to the medial thigh with an antalgic limping gait. He also had difficulty in weaning himself from the walker during ambulation. He was then referred to our hip clinic. Serial follow-up radiographs revealed only a minimal amount of callus formation at the fracture site with no sign of progressive healing in the past 3 months (Figure 3), which met the criteria for nonunion by the U.S. Food and Drug Administration (FDA) [7]. Computed tomography (CT) for the pelvis performed at 9 months postoperatively confirmed the nonunion status (Figure 4). Though we suggested surgical options such as revision reduction and fixation, and total hip replacement, the patient preferred nonoperative treatment to revision surgery. As a part of shared decision making, the patient then received subcutaneous teriparatide injections, 20 mcg daily. A follow-up CT, 2 weeks before completion of the 4th month of treatment, showed a united fracture (Figure 5). Though some femoral head osteonecrosis was also noted on the CT, which may be associated with the initial traumatic event, the patient reported improved left hip pain. He could ambulate independently and without aids afterward. We then discontinued teriparatide after the 4-month treatment and shifted to denosumab to continue the anti-osteoporosis treatment. Further observation was necessary to observe the progression of the femoral head osteonecrosis.

## 3. Discussion

This case suggests that teriparatide may be a potentially viable nonsurgical treatment for femoral basicervical fracture nonunion. Surgical treatment for basicervical fracture nonunion remains the mainstream approach. Patients may have the surgical treatment immediately after the diagnosis of nonunion or after expectant treatment. Only a few patients have achieved union status after prolonged observation [1,4,5]. There are potential complications following revision surgery, including an increased risk of neurovascular injury, surgical site infection, prolonged hospitalization and medical costs. For patients requiring autogenous bone grafting, donor site morbidity may also bring a negative impact for patients. Elderly patients are more susceptible to operative complications [4,8]. A conservative therapy to treat nonunion with minimal adverse effects would be of great interest. However, owing to limited evidence, we do not know how effective teriparatide is in treating basicervical nonunion. Patients should be made aware of potential treatment failure and the consequences, such as implant failure and persistent pain.

Teriparatide, a human recombinant parathyroid hormone (PTH) analogue, has primarily been used to treat osteoporosis. Recently, more and more evidence has shown that teriparatide could accelerate bone healing and may be an effective treatment for fracture nonunion [6,7]. Aspenberg et al. conducted a randomized, double-blind clinical trial and confirmed that teriparatide as an anabolic agent could accelerate distal radial fracture healing when compared with a placebo [9]. Coppola et al. reported four cases of nonunion after a lower limb fracture. They were treated with a teriparatide 20 mcg subcutaneous injection daily and adequate bone callus formation over the nonunion site was noted in all of the patients [10]. Yu et al. presented a middle-aged male with a femoral shaft fracture, whose initial nonunion had failed to heal one year after a second surgical intervention with autogenous bone grafting. Teriparatide was administered once daily for 9 months and follow-up radiographs showed bone bridging with a decreased fracture gap, suggesting fracture union. Furthermore, no side effects were observed [11]. The author stated that teriparatide may be a potentially superior choice to autogenous bone grafting in treating fracture nonunion. Other successful cases using teriparatide in healing different kinds of fracture nonunion have also been reported, including in the sternum, odontoid and ulna [12,13,14]. The literature focused on teriparatide treatment for proximal femoral fracture nonunion is relatively scanty. Yuichi et al. administered teriparatide once weekly to a rheumatoid patient with femoral neck fracture nonunion, and bone healing was confirmed by computed tomography with decreased ambulatory groin pain [15]. Lee et al. reported three cases of femoral fracture nonunion. All the patients received 20 mcg of teriparatide per day after nonunion was diagnosed. Bone union was observed on radiographs in all the patients after 3 months of medical treatment and no teriparatide-related side effects were noted [16]. In our case, bone union was achieved before completion of the 4-month treatment. No adverse events associated with teriparatide were observed.

## 4. Conclusions

Previous studies have suggested teriparatide to be a potentially viable treatment for numerous kinds of fracture nonunion. Our case goes a step further to expand its possible application for basicervical peritrochanteric fracture nonunion. The success rate, the optimal treatment duration and the comparability with surgical treatment remain unclear. Larger scale studies are necessary to further elucidate the efficacy and to build up a treatment protocol.

## Figures and Tables

**Figure 1 medicina-58-00983-f001:**
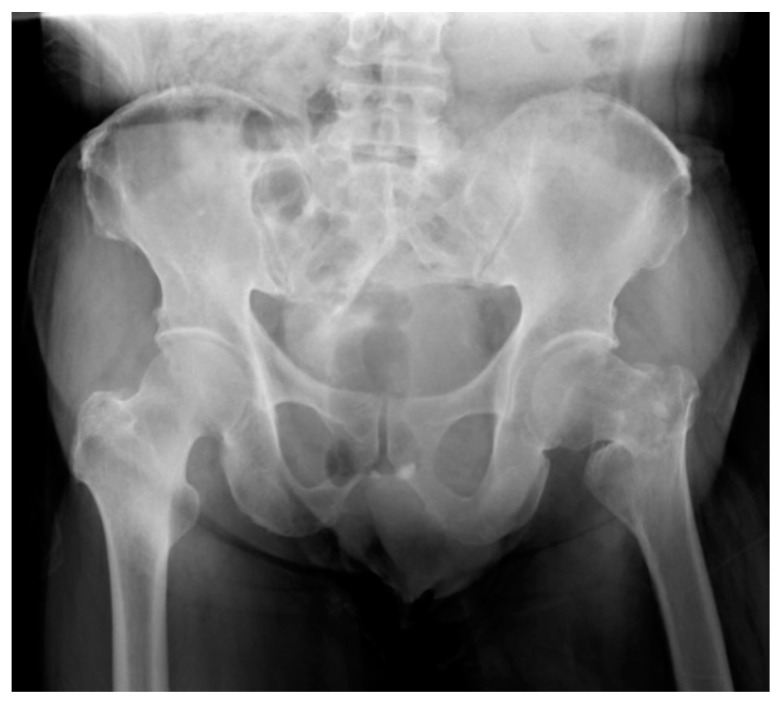
AP radiograph of the pelvis, showing left basicervical peritrochanteric fracture with displacement and varus angulation.

**Figure 2 medicina-58-00983-f002:**
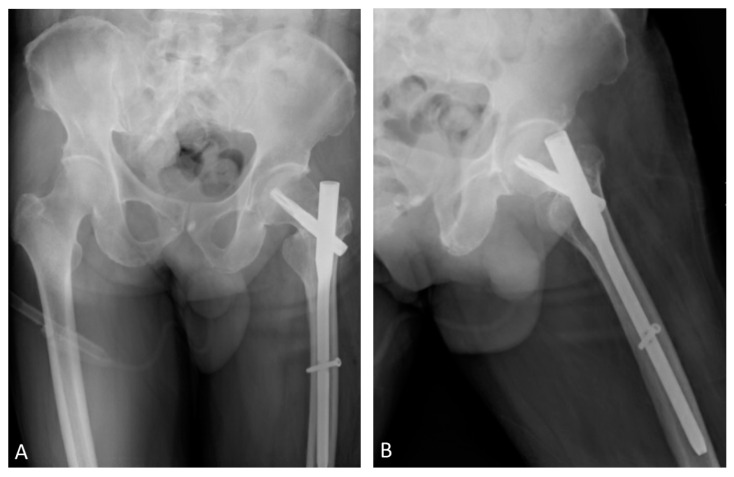
(**A**,**B**) Orthogonal views of postoperative radiographs, showing varus alignment and neck shortening.

**Figure 3 medicina-58-00983-f003:**
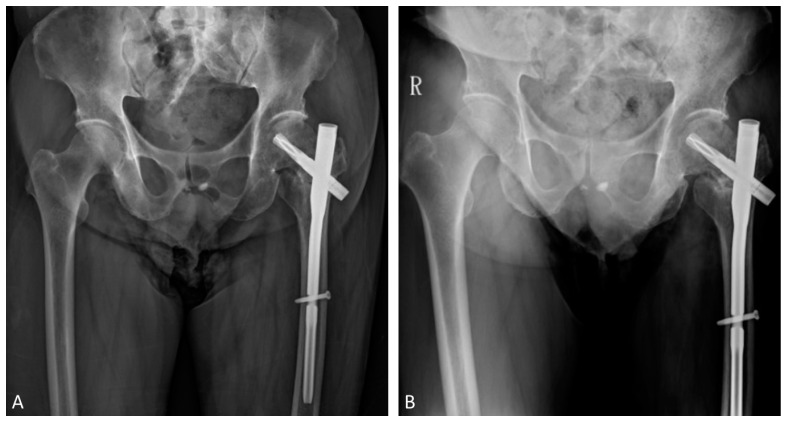
(**A**) 3 months (**B**) 6 months postoperative AP radiograph of the pelvis, revealing only minimal amount of callus formation. ”R” in the figure stands for “right,” which is the by default marking of laterality for plain films in the institute.

**Figure 4 medicina-58-00983-f004:**
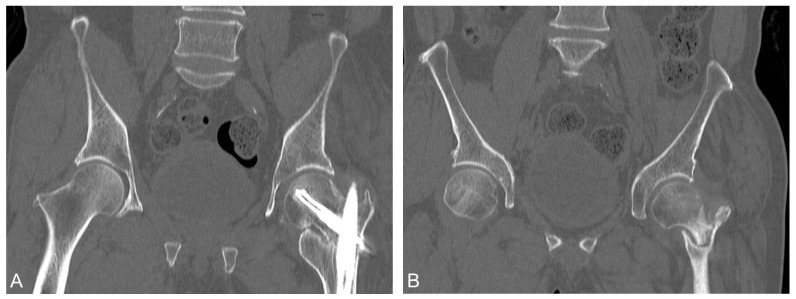
(**A**,**B**) CT performed at 9 months postoperatively. Fracture gap was still evident.

**Figure 5 medicina-58-00983-f005:**
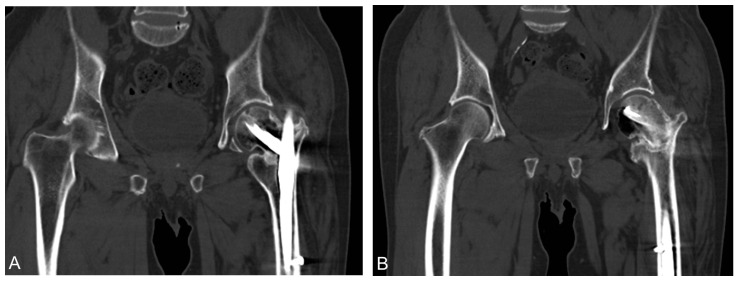
(**A**,**B**) CT performed before completion of 4-month teriparatide treatment showed fracture union with bridging callus formation.

## Data Availability

Not applicable.

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
