# Peer review of "Teriparatide as an Effective Nonsurgical Treatment for a Patient with Basicervical Peritrochanteric Fracture Nonunion—A Case Report"

_medicina, 2022, doi:10.3390/medicina58080983_

Round 1
Reviewer 1 Report
First of all, congratulations on the successful operation of a special difficult case.
The overall design of the article is well thought out.
1) In line 70, it is expressed as minimal amout calluus formation.
What criteria did you use when defining nonunion? Please clarify the definition applied to the patient.
2) In line92, "alternative" is a hasty and not appropriate expression.
3) What I am curious about is, are there any revision cases where the authors are not unionized even when teriparatide is used in similar cases? I believe that the author and other authors have had similar experiences. Therefore, I think that the above case is luckily unioned.
Readers should not be confused.
In spite of using teriparatide, please present the case where there is no union and the reason, and present the difference between the union case and non-union case in the discussion.
Author Response
For Reviewer 1
Thank you for the thorough review and the constructive comments. The following are the point-by-point response.
- In line 70, it is expressed as minimal amout calluus formation.
Thank you for pointing out the Typo. They are corrected to be “amount of callus…”
- What criteria did you use when defining nonunion? Please clarify the definition applied to the patient.
We applied the FDA criteria, “A fracture that persists for a minimum of 9 months without signs of healing for three months.” The description was added, as well as a new reference 7, in the revised manuscript. We also added it to the abstract and made some rephrasing.
- In line92, "alternative" is a hasty and not appropriate expression.
Yes, thank you for the kind minder. We rephrased it to be “mat be a potentially viable…” for better expression.
4) What I am curious about is, are there any revision cases where the authors are not unionized even when teriparatide is used in similar cases? I believe that the author and other authors have had similar experiences. Therefore, I think that the above case is luckily unioned.
Readers should not be confused.
In spite of using teriparatide, please present the case where there is no union and the reason, and present the difference between the union case and non-union case in the discussion.
Reply:
This is a very important viewpoint. Thank you for helping us to further clarify the situation.
Actually, this is the first case that we used teriparatide to treat basicervical nonunion, owing to scarce evidence. We nearly always do revision surgeries for the patients with basicervical nonunion, whether immediately after diagnosis or after expectant treatment. Only few patients achieved union status after prolonged observation. Treatment with teriparatide for the case is a result of shared decision making. The patient did not feel like having another surgical treatment despite thorough discussion. Luckily bone union was achieved.
In the revised manuscript, we emphasized more about the consequence of basicervical nonunion, and that surgical treatment is usually more appropriate. We also rephrased the sentences to clarify that teriparatide is only a potentially viable treatment. Patients should be aware of the consequence of persistent nonunion, such as persistent pain and implant failure.
Furthermore, the regulations of reimbursement for teriparatide in Taiwan, set by the National Health Insurance Administration, is very strict. Besides, treating nonunion with teriparatide is off-label use. Therefore, patients need to pay out of pocket for the relatively expensive medication in this situation. We cautiously choose teriparatide as nonoperative treatment more often for scenarios with more solid evidence, such as stable fragility fracture of pelvis, atypical fractures, or certain other types of fracture nonunion. That may explain why this is our first case treated with teriparatide for basicervical nonunion.
Reviewer 2 Report
Thank you for the opportunity to review the above paper entitled "Teriparatide as an Effective Nonsurgical Treatment for a Patient with Basicervical Peritrochanteric Fracture Nonunion - a Case Report".
The authors present a case of a proximal femoral fracture non-union at nine months post-operatively in a 70 year-old male patient who was treated by daily administration of subcutaneous teriparatide injection for four months. They provide sufficient evidence for the treatment protocol and its effectiveness.
It is a well written paper and adds in the existing literature as similar cases are rare and randomized control studies are difficult to be done.
Although the use of teriparatide in management of non-unions is promising, one main issue is the cost, especially when it is used off-label. In many countries teriparatide is an expensive drug and is not covered by insurance companies for indications other than severe osteoporosis. How did the authors covered the cost for their patient? Do they have any suggestions how to overcome the issue of high cost in order to use teriparatide widely in the treatment of fractures or non-unions? Please comment.
Author Response
For Reviewer 2
Thank you for the thorough and the insightful review. The followings are the point-by-point response.
- Although the use of teriparatide in management of non-unions is promising, one main issue is the cost, especially when it is used off-label. In many countries teriparatide is an expensive drug and is not covered by insurance companies for indications other than severe osteoporosis. How did the authors covered the cost for their patient? Do they have any suggestions how to overcome the issue of high cost in order to use teriparatide widely in the treatment of fractures or non-unions? Please comment.
Reply:
The regulations of reimbursement for teriparatide in Taiwan, set by the National Health Insurance Administration, is very strict, as a second-line anti-osteoporosis medication. As you have mentioned, treating nonunion with teriparatide is off-label. Therefore, patients need to pay out of pocket for the relatively expensive medication in this situation. We cautiously choose teriparatide as nonoperative treatment more often for scenarios with more solid evidence, such as stable fragility fracture of pelvis, atypical fractures, or certain other types of fracture nonunion.
This is the first case that we used teriparatide to treat the basicervical nonunion, owing to scarce available evidence. It is a result of shared decision making because the patient did not feel like having another surgical treatment despite thorough discussion.
Surgical treatment should still be the mainstream for treatment of basicervical nonunion. Only for certain clinical scenarios, e.g., severely ill patients not suitable for further surgical treatment, teriparatide may be a potentially viable treatment option. Achieving bone union is not guaranteed with current evidence. Patients should be aware of the potential consequences of persistent nonunion, such as implant failure and persistent pain. We would suggest that teriparatide as conservative treatment should only be applied after thorough discussion. Further studies of larger scale are necessary to provide more evidence. Only after more solid evidence are available may the associated health insurance covers its use for nonunion. The comments are added with modification in the discussion and some sentence in the abstract was also rephrased.
Thank you again.
Reviewer 3 Report
Originality: This is a very short paper, providing a minimum of information about a case study, and constitutes a novel contribution to the literature focusing on teriparatide treatment for proximal femoral fracture nonunion were scanty.
Now, in the introduction section is necessary expand the importance and state of art related with this orthopedic problem.
Methodologically Sound: As a case study report it is rather hard to go wrong methodologically, and the paper conforms to the standard.
Follows Appropriate Ethical Guidelines: Whilst there is no obvious declaration of ethical approval, it would appear to be a report of actions taken as part of normal clinical practice (as a case study report), and thus is acceptable.
Has results which are clearly presented and support the conclusions: Again, it conforms to the usual format for the presentation of a case study, although the content is very sparce. It is, however, appropriate enough, and does report a rare case likely to be of interest to a healthcare audience.
Overall Scientific Quality: As a minor case study report it lacks scientific depth, but effectovely is intended only to report the occurence of a rare case and to highlight the importance of correct disgnosis, and on these grounds merits attention.
Correctly References Previous Relevant Work: It appears to reference prior work succinctly and accurately.
Importance/Interest: Although marked by its brevity, the content is of interest, particularly to clinicians such as orthopaedic who examine this conditions all the time, who may need to be aware of the variant forms of this orthopeadic problem.
Author Response
For Reviewer 3
Thank you for the thorough and constructive comments. The followings are the point-to-point reply.
Originality: This is a very short paper, providing a minimum of information about a case study, and constitutes a novel contribution to the literature focusing on teriparatide treatment for proximal femoral fracture nonunion were scanty.
Now, in the introduction section is necessary expand the importance and state of art related with this orthopedic problem.
Thank you for the perspectives. We add another systemic review about teriparatide treatment for fracture nonunion as reference 6. We also add few lines to describe that the available evidence for its application to basicervical fracture nonunion remained scarce in the introduction.
Methodologically Sound: As a case study report it is rather hard to go wrong methodologically, and the paper conforms to the standard.
Thank you for the review opinions.
Follows Appropriate Ethical Guidelines: Whilst there is no obvious declaration of ethical approval, it would appear to be a report of actions taken as part of normal clinical practice (as a case study report), and thus is acceptable.
Thank you for the reminder. The ethic statements are at line 145-6.
Has results which are clearly presented and support the conclusions: Again, it conforms to the usual format for the presentation of a case study, although the content is very sparce. It is, however, appropriate enough, and does report a rare case likely to be of interest to a healthcare audience.
Thank you for the review opinions.
Overall Scientific Quality: As a minor case study report it lacks scientific depth, but effectovely is intended only to report the occurence of a rare case and to highlight the importance of correct disgnosis, and on these grounds merits attention.
Correctly References Previous Relevant Work: It appears to reference prior work succinctly and accurately.
Thank you for the review opinions.
Importance/Interest: Although marked by its brevity, the content is of interest, particularly to clinicians such as orthopaedic who examine this conditions all the time, who may need to be aware of the variant forms of this orthopeadic problem.
Thank you for the review opinions.